# Programming molecular topologies from single-stranded nucleic acids

Xiaodong Qi [1,2], Fei Zhang[1,2], Zhaoming Su[3,4], Shuoxing Jiang [1,2], Dongran Han[5,6], Baoquan Ding[7,8], Yan Liu [1,2], Wah Chiu[3,4], Peng Yin [5,6] & Hao Yan[1,2]

Molecular knots represent one of the most extraordinary topological structures in biological polymers. Creating highly knotted nanostructures with well-defined and sophisticated geometries and topologies remains challenging. Here, we demonstrate a general strategy to design and construct highly knotted nucleic acid nanostructures, each weaved from a single-stranded DNA or RNA chain by hierarchical folding in a prescribed order. Sets of DNA and RNA knots of two- or three-dimensional shapes have been designed and constructed (ranging from 1700 to 7500 nucleotides), and they exhibit complex topological features, with high crossing numbers (from 9 up to 57). These single-stranded DNA/RNA knots can be replicated and amplified enzymatically in vitro and in vivo. This work establishes a general platform for constructing nucleic acid nanostructures with complex molecular topologies.

[1] School of Molecular Sciences, Arizona State University, Tempe, AZ 85287, USA. [2] Center for Molecular Design and Biomimetics, Biodesign Institute, Arizona State University, Tempe, AZ 85287, USA. [3] Department of Bioengineering and James H. Clark Center, Stanford University, Stanford, CA 94305, USA. [4] Division of Cryo-EM and Bioimaging, SSRL, SLAC National Accelerator Laboratory, Menlo Park, CA 94025, USA. [5] Department of Systems Biology, Harvard Medical School, Boston, MA 02115, USA. [6] Wyss Institute for Biologically Inspired Engineering, Harvard University, Boston, MA 02115, USA. [7] CAS Key Laboratory of Nanosystem and Hierarchical Fabrication, CAS Center for Excellence in Nanoscience, National Center for Nanoscience and Technology, China, Beijing 100190, China. [8] University of Chinese Academy of Sciences, Beijing 100049, China. These authors contributed equally: Xiaodong Qi, Fei Zhang, Zhaoming Su. Correspondence and requests for materials should be addressed to F.Z. (email: fei.zhang@asu.edu) or to H.Y. (email: hao.yan@asu.edu)

Naturally existing molecular knots have been found in biopolymers, such as DNA and proteins. During protein folding, the protein molecules occasionally exhibit small amounts of crossings[1–4]. Knotted DNA structures are usually formed during genomic DNA replication and transcription and are eventually resolved by DNA topoisomerases[5]. Although knotted DNA structures are present in some bacterial phage genomes during the packing of the genomic DNA into the phage capsid[6,7], they are not very predictable or well-organized. The construction of molecular knots requires a high level of control over their assembly behaviors at a nanometer scale[8]. It is challenging to design and construct highly knotted nanostructures with well-defined topologies and geometries. The programmability of nucleic acids[9,10] makes them excellent candidates to use for creating topological knots at a molecular level. Synthetic topological DNA nanostructures have previously been constructed by creating topological nodes based on B-form/Z-form double-stranded DNA (dsDNA) helices[11–14], paranemic crossovers[15,16], and DNA four-way junctions[17]. In contrast to the strategies that rely on the design of individual topological nodes and finding the appropriate DNA motifs/interactions with which to assemble such nodes, a different approach is to fold one long single-stranded DNA (ssDNA) chain into programmable topologies. However, this strategy presents an important yet unanswered question: can partially paired, dsDNA thread through its own entanglement to form highly knotted topological structures?

Here, we present a general approach to creating topological knots of ssDNA and ssRNA with high crossing numbers (up to 57). A crossing number is a knot invariant that shows the smallest number of crossings in any diagram of the knot, representing the topological complexity of a knot[18]. Each of these ssDNA or ssRNA knots are folded and then self-assembled from a replicable single-stranded nucleic acid molecule with sizes ranging from 1700 to 7500 nucleotides (nt). Various ssDNA knots that had different crossing numbers and that displayed two-dimensional (2D) patterns were designed, constructed, and characterized using atomic force microscope (AFM) imaging. The yield of the folded knots was optimized by programming a step-wise hierarchical folding pathway through sequence design in the paranemic cohesion regions. A series of design rules have been formulated to significantly improve the yield of well-formed target structures. Furthermore, we expanded the design strategy to construct ssRNA knots and three-dimensional (3D) ssDNA knots. The 3D knots were characterized by cryogenic transmission electronic microscopy (cryo-EM) single particle reconstruction, which confirmed their designated geometries.

Our ability to construct knotted structures that can be replicated and transcribed enzymatically, with the use of designer single-stranded nucleic acids that can self-fold into molecular knots of customized shapes, allows us to significantly reduce the cost of production as compared with the cost of the multi-stranded DNA nanostructure systems. The single-stranded topology may also open up more opportunities for us to select and generate nanostructures with desired functions by using in vitro evolution. In addition, our work establishes a fundamental and general platform for constructing nucleic acid nanostructures with increased sizes and unprecedented complex molecular topologies. This ability to construct such structures may allow us to create more intricate diverse transformative applications in nanotechnology and molecular science, such as nanophotonics[19,20], drug delivery[21], cryo-EM analysis[22], and DNA-based memory storage[23,24].

## Results

**Design principles of ssDNA or RNA knots**. To create ssDNA knots, we used DNA parallel crossover (PX) motifs as the modular building blocks and a node-edge network as the geometric blueprint for arbitrary nanostructures, as they can be readily connected with other PX motifs to enable single-stranded routing. In comparison with the compact parallel or antiparallel helical arrangements, wireframe networks are better candidates for constructing knotted structures, as they offer more space for DNA chains to thread through during the early formation of partial structures. For example, knot $9_1$ (Alexander–Briggs notation) can be assembled by either connecting nine right-handed X-shaped junction tiles together (Fig. 1a) or by threading a single chain through itself nine times (Fig. 1b). Here, we used two parallel crossovers that were separated by four or six base pairs to form an X-shaped topology, which represented one cross node in a knot (Fig. 1c). We arranged the 9 × X-shaped DNA tiles into a square with 2, 2, 2, and 3 crossings on the four edges, respectively (Fig. 1d), and separated the adjacent PX junctions by one turn (10 or 11 bp) or two turns (21 bp) of the dsDNA. After connecting the nearest DNA strands and adding small linking structures at the four vertexes, the resulting design consisted of only one long ssDNA (Supplementary Figure 1 and Fig. 1d). The overall routing of the ssDNA can be treated as a two-step process. First, half of the DNA chain folded back to partially pair with the other half of the DNA, leaving several unpaired single-stranded regions (~4–6 nts) in between the perfectly paired regions (~10, 11, or 21 bps). Then, the unpaired regions matched to each other by paranemic cohesion interactions and finally knotted into the target topology (Fig. 1d).

Designing a topologically and kinetically favorable folding pathway is a key step for the successful formation of intricate structures with high crossing numbers. We introduced a hierarchical folding strategy to guide the knotting process in a prescribed order. For example, a knot with 23 crossings could be assigned to a location on a three-column grid that is represented by a rectangle with three square cavities (Fig. 2c). To maintain the structural stability and rigidity of each edge, we limited the length and number of crossings on each edge: in any six turns (63 bp) of a double helical DNA, only two or three PX crossings were allowed. In total, 23 crossings were assigned on the 10 edges, in which seven edges had two crossings and three edges had three crossings (Fig. 2c). The order of the folding pathway could be designed in many possible ways (Supplementary Figure 2). We listed all of the possible combinations for the formation order of the crossings in the knot and compared their routing pathways (Supplementary Figure 2).

Three essential rules were identified for optimizing the folding pathway. First, a linear folding path is better than a branched one, because the linear folding pathways involve two free ends that thread to form the loops in a sequentially ordered pathway, while the branched folding pathways have parallel steps that each involves a single free end to thread through the preformed loops. Based on an entropic point of view, the formation of two free ends looping with each other is expected to be possibly easier than one free end threading itself through preformed loops (Supplementary Figure 3), agreeing with the previously reported simulation theory[25]. Second, in the early stage of folding when the unfolded portion of the strand is still long, the folding pathway should avoid threading DNA strand through any of its own preformed structures (Supplementary Figure 4). Third, the edges with three crossings should generally fold before the edges with two crossings, since the cohesion force provided by three paranemic interactions (totally 12–18 bp) is expected to be stronger than that from two paranemic interactions (totally 8–12 bp) with random sequence designs. If a two-crossing edge needs to be formed before three-crossing edges, the possible length design of the paranemic cohesion region is 12 base pairs (we will further discuss the sequence assignment in the following text). From the crossing positions and the grid layouts, the route for

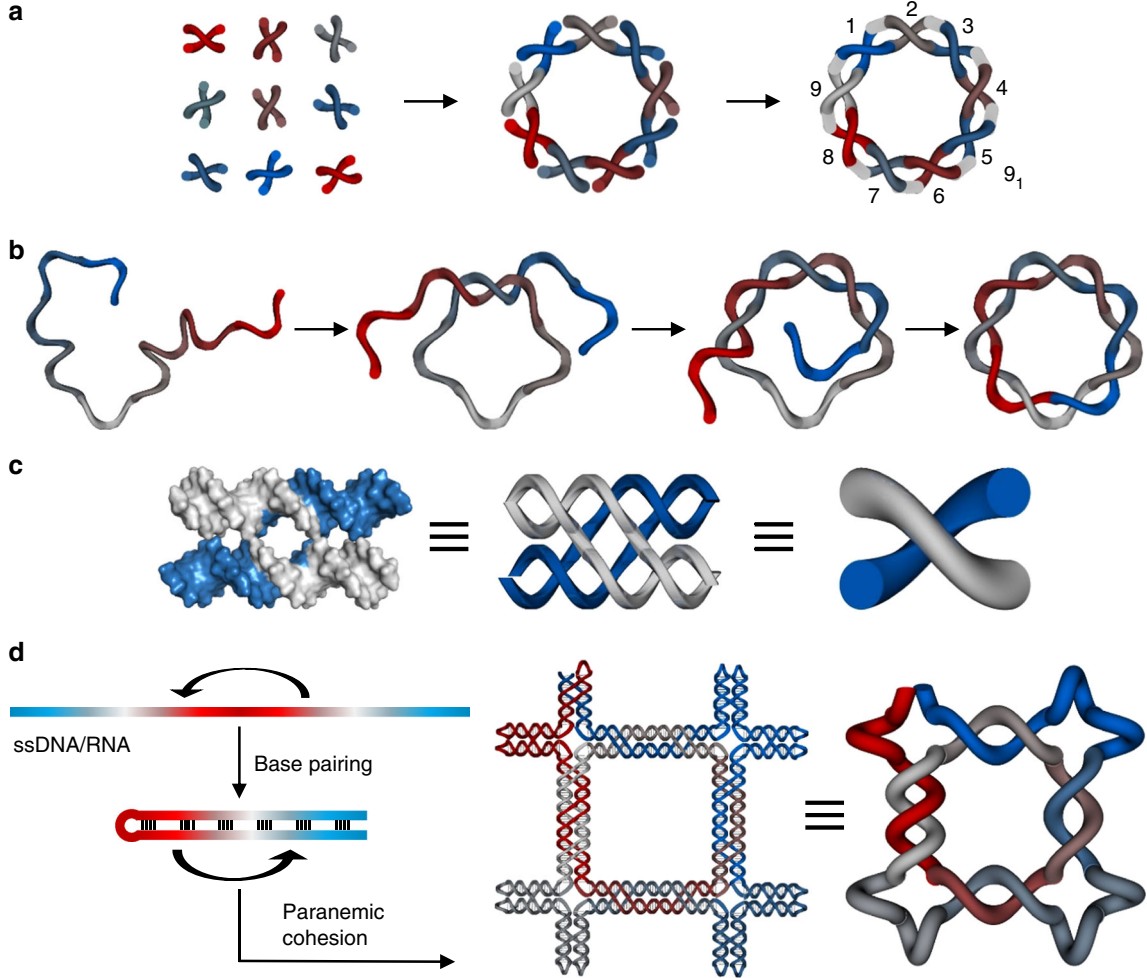

**Fig. 1** Design of single-stranded DNA (ssDNA) or RNA knots. A knot with a crossing number of nine can be constructed via two strategies: **a** Assemble with preformed individual X-nodes that are linked together with specific sticky ends associations and ligation. **b** Thread and knot a single chain into the target topology. The color scale shows different regions of a single chain. **c** Paranemic crossover motifs were introduced as the building blocks for the knotted nucleic acid nanostructures. **d** A schematic diagram that shows the design and folding pathway of a ssDNA to form knot $9_1$ as an example (The color scale shows different regions of ssDNA/RNA). A single-chain DNA was assigned with partially paired regions to first form a large loop-stem hairpin structure. Then, the unpaired loop regions were designed to interact with each other through paranemic cohesions to form the target knot. The formation of the knot involved the threading of the two ends of the loop-stem structure by following a pathway similar to that shown in **b**

the chain threading is determined by specifying the order in which the chain visits each vertex and knots on each edge. Figure 3a, Supplementary Figure 4, and Supplementary Table 1 show the selected folding pathway for the three-column grid knots. We compared the folding yield of the two types of scaffold routings by AFM imaging (Supplementary Figure 5). The selected linear folding pathway (Supplementary Figure 4a) produced 57.9% ($N = 214$) well-formed structures, while the branched one (Supplementary Figure 4b) showed a yield as low as 0.9% ($N = 221$).

The next step in our design procedure was to assign an appropriate sequence to enable the long ssDNA to create the structural and topological complexity. We established several criteria for generating a valid raw sequence: First, the ideal percentage of GC content in all regions of the DNA sequences was determined to be between 30 and 70%, since any GC content outside of this range would adversely affect the DNA synthesis. Second, depending on the size of the ssDNA, every segment that was 6–8 bases long was treated as one unit, to help us evaluate the uniqueness of the DNA sequence. The specificity of recognition between the designed base pairings rely on the uniqueness of the DNA sequences. Third, the repeating length of G was limited to 4

nt. A raw sequence was obtained by using the inherent algorithm of the Tiamat software[26], and in adherence with these rules. Then, the raw sequence was inspected manually and several modifications were made: The local sequences that were used to form the paranemic crossovers were checked to make sure that each of the crossovers were stable; the GC content in each of the paranemic cohesion regions were designed individually and interdependently as they needed to be compared with one another. It was necessary that all of the paranemic cohesions would have a sequentially decreasing melting temperature, ordered according to the predetermined folding pathway. Lastly, the uniqueness of the paranemic cohesions was optimized independently, such that mismatches and cross-talking in the second step of the folding were minimized.

**Synthesis of long ssDNA molecules**. Both the chemical and enzymatic synthesis of long ssDNA molecules are technically challenging, because the chain possesses a large portion of self-complementarity. As shown in the folding pathway, the ssDNA molecule will first form a long hairpin-loop structure with

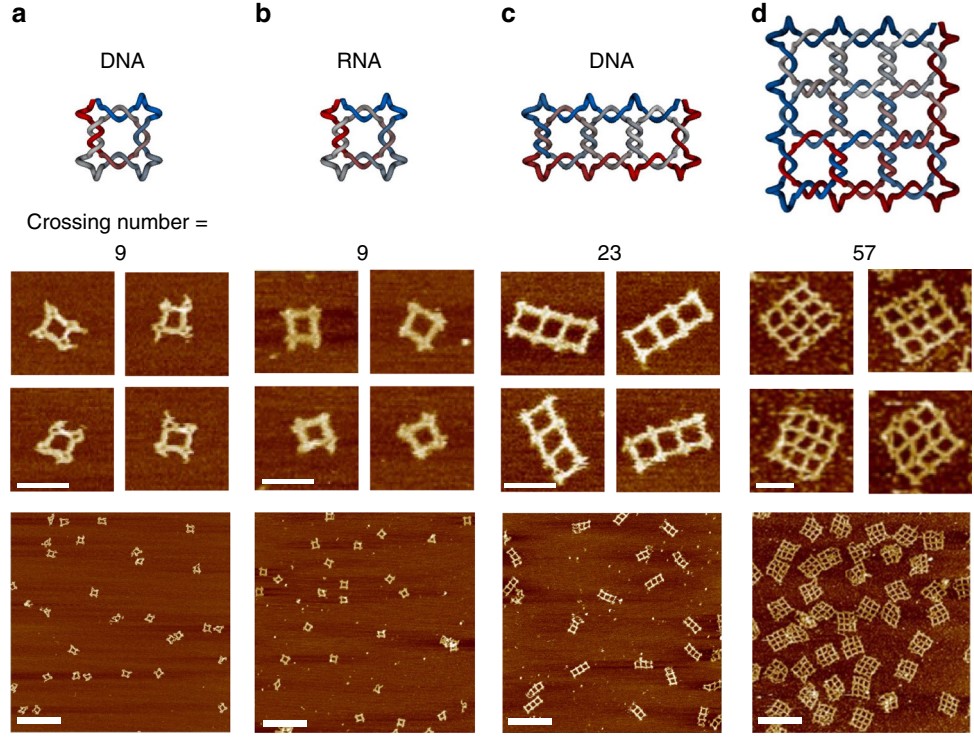

**Fig. 2** Design and AFM characterization of two-dimensional single-stranded DNA and RNA knots. Designer models (top row) for the 2D nanostructures and their corresponding AFM images (middle row shows the zoomed-in images and bottom row shows the zoomed-out images) with increasing crossing numbers: **a** A DNA square with a crossing number of 9, **b** An RNA square with a crossing number of 9, **c** A DNA rectangle with three square cavities and a crossing number of 23, **d** A DNA 3 × 3 square lattice with a crossing number of 57. The color scale in the schematics indicate the routing of the large long-stem structures. The scale bars in the zoom-in images represent 50 nm, while the ones in the zoom-out images are 200 nm

the 5' and 3' ends meeting each other. We first split the full-length ssDNA strand into two equal halves, with each strand lacking significant secondary structures, we then inserted each of them into plasmids as double-stranded genes, and then amplified them by cloning. The two dsDNA genes were obtained separately from the plasmids by restriction enzymes digestion (EcoRI + XbaI and XbaI + HindIII, respectively) and were then ligated together with a linearized phagemid vector, pGEM-7zf(-) (Supplementary Figure 6a). In order to obtain the full-length ssDNA molecule, the recombinant M13 phage was replicated in *E. coli* with the assistance of a helper plasmid, pSB4423[27]. Because the helper plasmid, pSB4423, does not contain a phage replication origin, only the phagemid vector containing the full-length ssDNA origami gene was able to act as a template for the phage DNA replication. After the extraction and purification of the recombinant phage DNA, EcoRV digestion was performed to cut out the target ssDNA (Supplementary Figure 6a). Native agarose gel electrophoresis was used to separate the target ssDNA from the phagemid vector ssDNA (Supplementary Figure 6b). Using this method, we synthesized and amplified all of the long ssDNA strands at a nanomole quantity (with 1 L scale of *E. coli* culture) and high purity. The ssDNA strand was obtained, then self-assembled (folded) in a 1x TAE-Mg buffer with a 12 h or 24 h annealing ramp from 65 °C to 25 °C (see details in Methods section). The folded products were then characterized by using AFM imaging, gel electrophoresis and/or cryo-EM imaging.

**Complex knots with large crossing numbers**. We applied our design procedures to create more complex DNA knots with increasing crossing numbers. A 3 by 3 square grid of DNA knots with 57 crossed nodes was designed with an optimized linear

folding pathway (Fig. 2d and Supplementary Figure 7). Other geometric layouts were also used following the same design principles. A large molecular knot with 67 crossings in a hexagonal lattice was designed and constructed. High-resolution AFM imaging was used to characterize the structural formations (Supplementary Figure 8). It is noted that the smaller knot structures with crossing numbers 9 and 23, folded well with yields as high as 69% ($N = 103$) and 58% ($N = 214$), respectively (Fig. 2a, c and Supplementary Figure 9 and 5). However, as the crossing number of the knot increased to 57 or 67, the folding yield dropped significantly and in the images, only 1.2% of the resulting structures were perfectly formed with the 57 crossings ($N = 327$) and none of the resulting structures were perfectly formed with the 67 crossings ($N = 389$) based on single-molecule analysis by AFM imaging (Fig. 2d and Supplementary Figures 7, 8). Almost every one of the formed structures we examined showed some degree of various folding defects (Supplementary Figure 8). With such a high complexity, even the hierarchical folding optimization did not significantly increase the overall yield (Supplementary Figure 10 and Supplementary Table 2). The folding behaviors in our ssDNA knots were remarkably different from that of the classic DNA structures. To make the target knots, the ssDNA chain needed to fold following an exactly defined order. If one crossing was misfolded in an earlier stage, it would be impossible (or at least extremely difficult) for it to correct itself afterwards. Nevertheless, the yields of those knots were not surprisingly low when compared with the yields of the chemical synthesis reactions that contained multiple steps. If we treated the formation of one crossing as one knotting step, the average yield for each knotting step could be estimated to be at least 90%, and

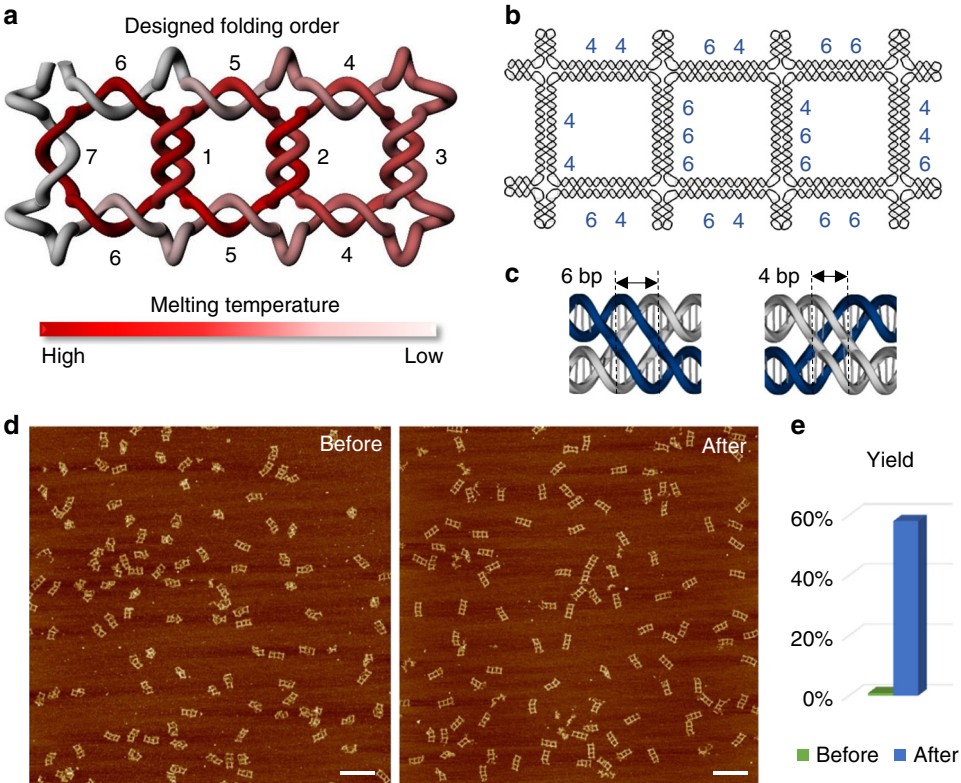

**Fig. 3** Optimization of the folding pathway for ssDNA knots. **a** A designed model shows the selected best folding pathway (from one to seven) for the three-square structure with 23 crossings, by following our optimization rules. The red to gray color scale as well as the number one to seven represent the order of the paranemic cohesion interaction strengths on the edges from high to low (one to seven), based on the number and length of the paranemic cohesions involved as shown in **b**. **c** The paranemic interaction regions can be designed with lengths of 4 bp or 6 bp with distinct expected binding strengths (6 bp > 4 bp). Therefore, one is able to guide the folding order of the knot structure by controlling the sequences and lengths of the paranemic interactions in each individual edge. We compared the folding efficiency of the known structures by using different folding pathways before and after optimization (**d**). The AFM images revealed a dramatic increase in the folding yield of well-form structures from 0.9% ($N = 221$) to 57.9% ($N = 214$) (**e**). The scale bars are 200 nm

in the low crossing number cases, the single step yield was as high as ~96%.

**Topological control to validate the knotting configuration.** As most of our high crossing number ssDNA knots were characterized by high-resolution AFM imaging (Fig. 2), one question was raised: whether the formation of the defected nodes in the final knot structure could be truly and completely identified with AFM imaging. We designed a link structure with eight nodes as a topological control (Supplementary Figure 11). This link structure contained two dsDNA rings (each only partially complementary), which connected to each other through eight paranemic cohesions. We constructed this particular structure by annealing two linear dsDNAs (each preformed from two ssDNAs) with eight stretches of mismatches (bubbles), each consists 6 nt (Supplementary Figure 11). The mismatches within these two linear dsDNAs would interact with their counterparts to form the stable paranemic cohesions. Nine-basepair sticky ends that extended from both ends of the dsDNAs closed the two rings after the formation of the correct structure. This link structure assembled well, as characterized by high-resolution AFM imaging (Supplementary Figure 11a). On the contrary, if the two linear dsDNAs were first ligated to form closed ring structures, we reasoned that these two dsDNA rings would not be able to assemble into the desired fully inter-locked loop structure. As expected, although the two dsDNA rings could still bind with each other partially through some of the paranemic cohesion

interactions (Supplementary Figure 11b), extensive defects were observed in all of the structures and unknotted structures can easily be scratched/deformed by AFM tips during scanning (Fig. 3, Supplementary Figure 11, and Supplementary Table 3), indicating the intact structures shown in Fig. 3a were knotted structures following the design.

**Design and construction of ssRNA knots.** Our design strategies for ssDNA knots can be adapted to create ssRNA knots. Although knots do not exist in all of the naturally occurring RNA structures discovered to date, they might be important factors in the early stages of evolution on earth, because knotted entanglements may be capable of conserving the spatial information of the RNA network, without involving covalent bonds in a harsh environment[28]. We designed an X-shaped RNA modular building block, which was similar to the PX structures of DNA. We followed the same steps for constructing the ssDNA knots. First, based on the 3D modeling of an A-form dsRNA helix (11 bp per helical turn, 19 degree inclination of base pairs) and the best geometric fitting, 8 (instead of 4 or 6) bp was chosen for the length of a paranemic crossover (Supplementary Figure 12). For an eight base-pair paranemic cohesion, a total of $4^8 = 65536$ possible sequences provided an adequate sequence space for the selection of unique complementarity to sufficiently avoid undesired interactions between the PX motifs. Second, given the 11 base pairs per turn of an A-form dsRNA, we assigned the lengths of the inter-motif stems as alternating between 8 and 9 bp (Supplementary

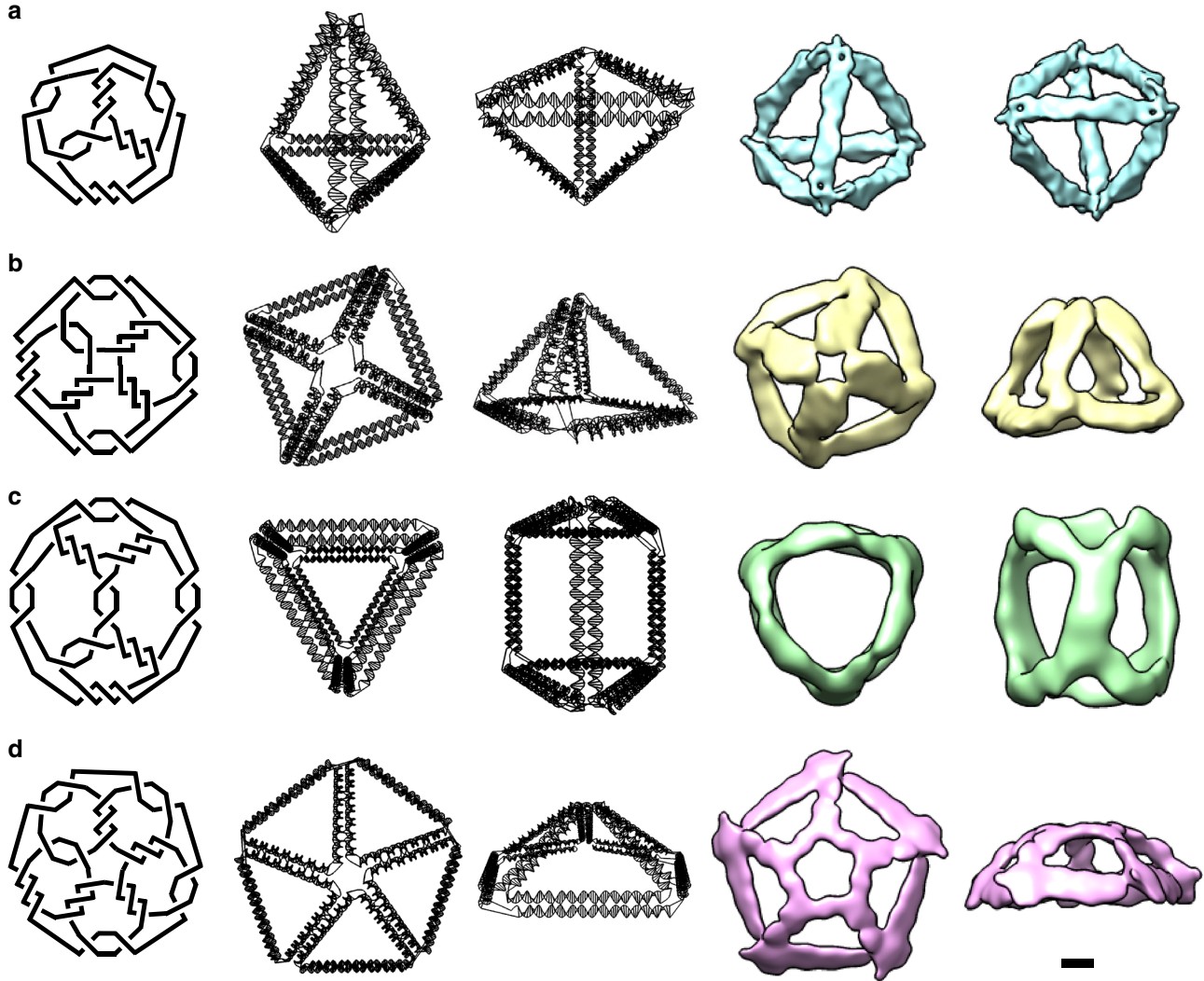

**Fig. 4** Design topologies and cryo-EM reconstruction of three-dimensional single-stranded DNA knots. A Schlegel diagram (left panel) transforms a three-dimensional object into a two-dimensional diagram. By following the same design rules of two-dimensional knots, several three-dimensional DNA knot frameworks were designed (middle panels of two different view angles for each structure). The cryo-EM reconstruction (right panel) revealed the correct formations of a tetrahedron with crossing number 15 (**a**), a square pyramid with crossing number 20 (**b**), a triangular prism with crossing number 22 (**c**), and a pentagonal pyramid with crossing number 25 (**d**). The scale bar represents 50 Å

Figure 12) to achieve a structural repeating unit of 33 bps for three full helical turns (i.e., 8 bp PX + 8 bp stem + 8 bp PX + 9 bp stem = 33 bp = 3 full turns). In this design, the neighboring structural units were in line with each other without accumulating helical twist, and the final assembled structure was expected to stay in 2D. Like the ssDNA $9_1$ knot, we assigned 2, 2, 2, and 3 crossings on the four edges of a square and looped the vertexes to form one single-stranded RNA (Fig. 2b). After generating the appropriate sequences by following the same sequence design rules as the ssDNA structures, the dsDNA gene coding for the long RNA strand was first synthesized, and then the ssRNA molecule was obtained by an in vitro transcription reaction. After annealing, the AFM images revealed the successful formation of the ssRNA $9_1$ knots (Fig. 2b and Supplementary Figure 12).

**Producing 3D ssDNA knots.** The design procedures presented here can also be applied to create 3D architectures with arbitrary geometries. We demonstrated the versatility of our method by constructing four ssDNA polyhedral meshes: a tetrahedron, a square pyramid, a triangular prism, and a pentagonal pyramid

with crossing numbers 15, 20, 22, and 25, respectively (Fig. 4). A Schlegel diagram was used to transfer the 3D objects to their topologically equivalent 2D nets. Optimized folding pathways were designed carefully for step-wise hierarchical assembly and the corresponding ssDNA strands were designed, synthesized, and assembled. AFM images showed an abundance of well-folded 3D nanoparticles with the expected sizes (Supplementary Figures 13–16 and Supplementary Tables 4–7). Single particle cryo-EM 3D reconstruction revealed that the overall conformations matched the designed geometries well (Fig. 4, Supplementary Figures 17, 18, and Supplementary Table 8). Notably, the vertex design for our ssDNA knots was different from the multi-arm junction design based on the double-crossover (DX) motif[29,30]. Instead, there is a 5 bp difference in length between the two parallel dsDNAs that form each edge, leading to chiral vertices and inclined edges in the ssDNA knots (Fig. 4). This unique geometric feature could be used to identify conformational diastereomers, which is when the structure is turned inside out, while satisfying all programmed Watson–Crick base pairing with the same network connectivity. The 3D reconstruction data

suggested that the ssDNA knots preferred to point the major grooves inwards at the vertices. A similar feature had been previously observed for the wireframe DNA nanostructures[31,32].

## Discussion

It has been a long-standing challenge to construct molecular knots with increasing size and complexity in a programmable and controllable way. By folding single-stranded nucleic acids with completely custom-designed sequences, we created ssDNA/ssRNA nanostructures with highly intricate topologies that were programmable, potentially replicable, and scalable. Compared with the unknotted DNA octahedron that was assembled from a 1.7 k base scaffold and a small number of auxiliary strands, that was reported in 2004[33], our ssDNA knots employed one single routing strand that had high crossing numbers without the help of any auxiliary DNA. Various 2D and 3D shapes have been designed and successfully constructed with a surprisingly high yield (~96%) of crossing steps. The same strategy has also been adapted for the design and construction of complex ssRNA knots. Compared with previously reported synthetic DNA topological structures[11–14,17,25,34–36], our ssDNA knots have the largest synthetic DNA knots (up to 7.5 k bases) and the most complicated topology (as high as 57 crossing number, Supplementary Figure 19). We also demonstrated that the hierarchical folding pathways significantly improved the folding yield. However, there are limitations of hierarchical folding as we will run out of choices for sequences with the increase of the crossing numbers.

In our previous reported ssDNA origami designs[37], we explored the unknotted 2D structures with zero crossing number. There are two major difference between current knotted ssDNA with the unknotted ones. First, we employed the wireframe/porous layout of DNA origami instead of highly packed version to provide sufficient space and allow the DNA strand to thread through itself. Second, we demonstrated several 3D structures of the knotted ssDNA here using Schlegel diagram to adapt our design method for 3D configurations. In addition, the knotted ssDNA origami strategy can be applied to form catenanes. For example, in the topological control experiment, the DNA structure assembled from two linear and partially paired dsDNAs can be ligated to seal the nicks (Supplementary Figure 11). The catanane structure is thus formed with two dsDNAs.

The single-stranded folding process of nucleic acids is a unique feature that can be used for the next-generation nucleic acids assemblies[38], and can be replicated and amplified in biological systems for cost-efficient, large-scale production. Although knots rarely occur in proteins and DNA, and RNA knots is still elusive up to date, it is believed that knots may play important roles in various biological systems and the evolutionary process, particularly during the origins of life[28]. The entanglement may provide extra stability to resist environmental temperature changes. More importantly, such structures have the potential to be used for programming nucleic acid synthesis in target cells to produce nanostructures, such as nanoscale devices, that may harness functions in vivo. One of the next research directions would be to transform the ssDNA/ssRNA nanostructures to be isothermal and dynamic in response to different stimuli in the cellular environment. A potential limitation of the current design for in vivo applications is the long-range interactions inside one single chain, which currently is difficult to achieve without annealing. This could be overcome by introducing localized interactions into the ssDNA/ssRNA design so that the local folding may become cotranscriptional and the global folding/threading occurs afterward. Another topic for further study is to understand the ordered folding process in the highly knotted structures, which is not only scientifically interesting, but also fundamentally important in understanding the mechanism of self-assembly in knotted bio-materials.

From a broader perspective, we hope that our approach for creating programmable ssDNA and ssRNA knots might enable the efficient evolution of functional molecular systems that resemble what might have occurred during the early stages of life on Earth. Programmable ssDNA and ssRNA knots also open up opportunities to engineer molecular devices and multivalent aptamer to adopt increasingly better performance through directed evolution.

## Methods

**DNA and RNA sequence design**. DNA/RNA structures and sequences were designed using the Tiamat software (Yanlab.asu.edu/Tiamat.exe)[26]. DNA and RNA sequences were generated by using the following criteria in the Tiamat software: (1) unique sequence limit: 8 nt; (2) repetition limit: 6–8 nt; (3) G repetition limit: 4 nt; (4) GC content: 0.45–0.55. Once sequences were generated, a few nucleotides were adjusted to eliminate the restriction enzyme targeting sequences (e.g., by EcoRI, EcoRV, HindIII, and XbaI) for cloning purposes. Then, the raw sequences of the paranemic cohesion regions were inspected manually and several modifications were made: the local sequences that were used to form the paranemic crossovers were checked to make sure that each of the crossovers were stable; the GC contents in each paranemic cohesion region were designed individually and interdependently as they needed to be compared with one another so that all of the paranemic cohesions would have a strength that was ordered sequentially according to a predetermined folding pathway; lastly, the uniqueness of the paranemic cohesions was optimized independently, so that the occurrence of mismatches and cross-talking in the second step of the folding process were minimized. For ssRNA origami sequences, a T7 promoter sequence was followed by two or three consecutive Gs that were manually incorporated onto the 5' end of the strand in order to facilitate efficient in vitro transcription reactions during gene synthesis.

**ssDNA and RNA synthesis**. The ssDNA origami sequences were divided into two fragments with restriction sites added onto both ends. The first fragment contained EcoRI and XbaI restriction sites and the second one contained XbaI and HindIII restriction sites. The EcoRV sites were also manually added to both ends of the full-length sequence in order to facilitate the production of the final ssDNA. The two DNA fragments that were ordered as double-stranded genes in plasmids were from Biobasic Company (Biobasic.com) with sequences that were verified through Sanger sequencing. The two DNA fragments that were cleaved from the plasmids by the restriction enzymes were accordingly subcloned into an EcoRI and a HindIII linearized pGEM-7zf(-) vector (Promega). After sequencing verification, the pGEM-7zf(-) vector that contained the full ssDNA genes was co-transformed into *E. coli* DH5α competent cells along with a helper plasmid pSB4423, a kind gift from Dr. Stanley Brown (Niels Bohr Institute, Denmark). *E. coli* colonies were formed after overnight incubation at 37 °C, and a single colony was inoculated into the 2xYT medium that had been supplemented with 2 mM MgSO$_4$ (Sigma-Aldrich) and grown at 37 °C overnight with shaking at 250 rpm. During the overnight growth, the recombinant M13 phages were continuously produced and secreted into the medium. During the next day, the culture was centrifuged at 5000 g for 15 min in order to pellet down the *E. coli* cells. The recombinant M13 phage was precipitated from the recovered supernatant with the addition of NaCl (to 30 g per liter) and PEG8000 (to 40 g per liter), and incubated in the 4 °C cold room for 1 h. The precipitated phage was then collected by centrifugation at 4500 g and 4 °C for 15 min and resuspended in the TE buffer (10 mM Tris-HCl pH 8.0 and 1 mM EDTA). The phage was lysed with two volumes of PPB2 buffer (0.2 M NaOH and 1% SDS), and mixed with 1.5 volumes of PPB3 buffer (3 M potassium acetate pH 5.5), and centrifuged at 5000 g for 15 min. The supernatant was then precipitated with one volume of 100% ethanol, and centrifuged for another 15 min at 5000 g. The phage DNA precipitate was then washed once with 75% ethanol and dissolved in the TE buffer. The phage ssDNA was digested by EcoRV enzyme (New England Biolabs) and resolved on a 1% agarose gel. The correct bands were sliced and purified by using a Monarch DNA Gel Extraction Kit (New England Biolabs).

For the ssRNA molecule synthesis, the DNA sequence with a T7 promoter at the 5' end was first cloned into a pUC19 vector by using the same method as was used for the ssDNA gene cloning process that was described above. The plasmid containing the ssRNA gene was linearized by using a HindIII enzyme (New England Biolabs) and the plasmid was purified by using a Phenol/chloroform extraction and ethanol precipitation. The in vitro transcription reaction was carried out by using the T7 RiboMAX Express Large Scale RNA Production System (Promega), following the manufacturer's instructions. The RNA molecules were then purified via a RNA Clean & Concentrator-25 kit (Zymo Research).

**ssDNA and RNA origami assembly**. The purified DNA and RNA molecules were diluted to 5–10 nM in 1× TAE-Mg buffer (40 mM Tris, 20 mM acetic acid, 2 mM EDTA, and 12.5 mM magnesium acetate, pH 8.0). The resulting solution was

annealed from 65 °C to 25 °C with a cooling ramp of 1 °C per 20 min to form the desired structures.

**AFM characterization.** All samples were imaged in "ScanAsyst mode in fluid," using a Dimension FastScan microscope with PEAKFORCE-HiRs-F-A tips (Bruker Corporation). After annealing, 2 μl of each sample was deposited onto a freshly cleaved mica surface (Ted Pella, Inc.), and left to adsorb for 1 minute. Then, 80 μl of 1× TAE-Mg buffer and 2 μl 100 mM of a NiCl$_2$ solution was added onto the mica, and 40 μl of the same buffer was deposited onto the microscope tip. The samples were then scanned by following the manufacturer's instructions.

**Topological control experiments.** The Biobasic company (Biobasic.com) synthesized four ssDNAs with the customized sequences and then cloned these sequences into a pBluescript SK(+) vector (Biobasic), with the gene sequences flanked by two BtsCI restriction sites. The final plasmids were then co-transformed with pSB4423 to produce recombinant M13 phages. The phage particles and phage DNAs were then purified by using the same methods as described in the ssDNA synthesis section. The ssDNAs were cleaved off from the recombinant phage DNAs by using a BtsCI restriction enzyme (New England Biolabs), and were gel purified by using a 4% urea denaturing PAGE gel. The four ssDNAs were annealed into two sets of dsDNAs (partially hybridized) in the 1x annealing buffer (50 mM Tris-HCl pH 8.0 and 100 mM NaCl). The two linear dsDNAs were mixed in a 20 nM concentration in a 1× TAE-Mg buffer, annealed from 65 °C to 25 °C at 1 °C per 20 min to form the desired paranemic cohesion interactions, and were then characterized by AFM imaging. The sticky ends on the two sets of the dsDNAs were able to close the ring structure without ligation. In the second control experiment, the two linear dsDNAs were ligated separately with T4 DNA ligase in a 1x ligation buffer (Thermo Fisher Scientific) at room temperature for 1 h to enable them to form the two circular dsDNAs. The ligation products were treated with exonuclease I and exonuclease III (New England Biolabs) to remove any linear DNA. A Monarch PCR & DNA Cleanup Kit (New England Biolabs) was used to further purify the solution. The two circular dsDNAs were then mixed at 20 nM in a 1× TAE-Mg buffer, annealed from 65 °C to 25 °C at 1 °C per 20 min and characterized by AFM imaging.

**Cryo-EM specimen preparation and data acquisition.** In the cryo-EM experiments, 2 μL of the aforementioned ssDNA nanostructure samples (concentrated to ~0.3 μM using Amicon 100 kDa centrifugal filters) were applied onto a 200 mesh R1.2/1.3 holey carbon Quantifoil grid (Quantifoil) that was cleaned with acetone (Sigma-Aldrich) for 12 h and glow discharged for 40 s before use. The grid was blotted for 3.5 s and immediately frozen in liquid ethane using a Vitrobot Mark IV (Thermo Fisher Scientific) with a constant temperature of 6 °C and humidity at 100%. The grid was stored in liquid nitrogen until the imaging session. All grids were examined on a JEM2200FS cryo-electron microscope (JEOL) that was operated under the following parameters: 200 kV, spot size 2, condenser aperture 70 μm, objective aperture 60 μm, 30,000 × magnification (corresponding to a calibrated sampling of 1.59 Å/pixel). The images were recorded under a low-dose condition on a direct detection device (DDD) (DE-20 4 k × 5 k camera, Direct Electron, LP) while operating in movie mode at a recording rate of 24 raw frames per second. The total dose was 40 electrons/Å$^2$ with the defocus ranging from 1.5 to 3 μm.

For the ssDNA tetrahedron sample, a total of 47 images were manually recorded on the DE-20 detector. Motion correction was performed by running the averages of three consecutive frames with the use of the DE_process_frames.py script (Direct Electron, LP). A total of 533 particle images were manually boxed, CTF corrected, and extracted in EMAN2.[39] Approximately 50 particles were used to generate a de novo initial model in EMAN2. The final 3D reconstruction with tetrahedron symmetry applied in EMAN2, resulted a cryo-EM density map at 17 Å resolution, calculated using the 0.143 criterion of the Fourier shell correlation (FSC) curve with a mask. A Gaussian low-pass filter was applied to the final 3D maps and displayed in the Chimera UCSF software package.[40]

Tilt-pair validation for the cryo-EM map was performed by collecting data at two goniometer angles, 0° and 10°, for each region of the grid. The test was performed using the e2tiltvalidate.py program in EMAN2. Additional details on the tilt-pair validation is provided in Table S1.

A total of 53, 75, and 46 micrographs of ssDNAs that folded into triangular prisms, square pyramids, and pentagonal pyramids were collected. Subsequently, 226, 330, and 238 particles were extracted, respectively. The initial models were generated as mentioned above and the final reconstructions were applied with the corresponding C3, C4, and C5 symmetries that yielded EM density maps at resolutions of 32, 25, and 26 Å. These resolutions were calculated using the 0.143 criterion of the Fourier shell correlation (FSC) curve with mask.

## Data availability

All data generated or analysed during this study are included in the paper and its Supplementary Information, and are available from the corresponding author on request.

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

## Acknowledgements

This research was supported by Office of Naval Research grant N000141512689 to H.Y. and National Science Foundation grants 1360635, 1563799, and 1334109 to H.Y. W.C. and Z.S. gratefully acknowledges funding support from National Institutes of Health (NIH P41GM103832 and P50GM103297). P. Y. acknowledges funding support from Office of Naval Research (N000141612410).

## Author contributions

F.Z. conceived the project. X.Q. and F.Z. designed the experiments. X.Q., Z.S., and S.J. performed the research. All of the authors analyzed the data. F.Z. and X.Q. wrote the paper. All the authors helped revise the paper.

## Additional information

**Competing interests:** . F.Z., H.Y. and X.Q. have filed a provisional US patent (patent number 62/663,678) based on this work. The remaining authors declare no competing interests.

