## [Peer Review File · Nature Communications]

Reviewers' Comments:

Reviewer #1:

Remarks to the Author:

Self-folding of single stranded DNA and RNA into defined geometrical nanostructures offers new ways to scale up nucleic acid assembly. Previously the authors demonstrated in a recent Science paper that single-stranded nucleic acids with thousands bases long can fold into origami nanostructures with 0 crossing numbers (therefore, no knots). In this paper, the authors demonstrated a novel hierarchical folding strategy to construct knotted nucleic acid nanostructures with high crossing numbers. The knotted structures demonstrated in this work exhibit unprecedented topological complexity, far beyond what has been achieved before using single stranded folding. Indeed, it is not only amazing but also surprising that the single stranded DNA and RNA can thread through its own chains and find a way to form such highly knotted structures, given the fact that the single strand has to weave through so many tangles. The authors showcased numerous examples of complex topology with high crossing numbers and demonstrated that they can program the formation of knots. Given the significant advancement from this work to design and construct unprecedented high complexity of knotted structures using programmable nucleic acid assembly and the convincing AFM, Cryo-EM and gel electrophoresis results, I recommend this manuscript for publication in Nature Communications.

Minor comments:

It is unclear from the description that kind of applications the highly knotted nucleic acid structures can offer? Can the authors elaborate their thoughts on the applications in the discussion? Could knotted structures provide more biological stability for future biomedicine applications?

It seems when the crossing number gets very high, the yield dropped significantly. This is understandable as the strand has to weave through so many cavities in its own tanglement. It is great to see the authors used hierarchical folding pathway to improve the yield, but can the authors comment on the limitation of hierarchical folding, given the fact that they will run out of choices for sequences?

Are the authors limited to only knotted structures? Is it possible to create other interesting topological structures, such as catenanes?

Reviewer #2:

Remarks to the Author:

The work by Qi et al. on "Programming Molecular Topologies from Single-stranded Nucleic Acids" demonstrates the design and experimental methods to form 2D and 3D nanostructures from a single strand of DNA or RNA. The work is closely related to recent work by the same labs published in 2017 (ref 37). The work is in itself impressive and well described, but the question is what advances have been achieved compared to the prior work on single stranded DNA nanostructures?

The two most significant differences are

- 1) Contrary to the former work the structures in the current MS contains knotted topological features. As a consequence, the folding pathway becomes highly important and it was demonstrated that good to high yields of folded structures are obtained by the right design.
- 2) The method was used to fold 3D wireframe structures whereas only 2D structures were formed in prior work.

These are significant and important advances and therefore I do believe that the work has sufficient impact to justify publication in Nature Comm. The manuscript is well-written and the work well-documented. I am however not sure if it is necessary to include the section on the synthesis of the long ssDNA/RNA unless it differs significantly from the method described in ref 37.

REVIEWERS' COMMENTS:

Reviewer #1 (Remarks to the Author):

Self-folding of single stranded DNA and RNA into defined geometrical nanostructures offers new ways to scale up nucleic acid assembly. Previously the authors demonstrated in a recent Science paper that single-stranded nucleic acids with thousands bases long can fold into origami nanostructures with 0 crossing numbers (therefore, no knots). In this paper, the authors demonstrated a novel hierarchical folding strategy to construct knotted nucleic acid nanostructures with high crossing numbers. The knotted structures demonstrated in this work exhibit unprecedented topological complexity, far beyond what has been achieved before using single stranded folding. Indeed, it is not only amazing but also surprising that the single stranded DNA and RNA can thread through its own chains and find a way to form such highly knotted structures, given the fact that the single strand has to weave through so many tangles. The authors showcased numerous examples of complex topology with high crossing numbers and demonstrated that they can program the formation of knots. Given the significant advancement from this work to design and construct unprecedented high complexity of knotted structures using programmable nucleic acid assembly and the convincing AFM, Cryo-EM and gel electrophoresis results, I recommend this manuscript for publication in Nature Communications.

Minor comments:

It is unclear from the description that kind of applications the highly knotted nucleic acid structures can offer? Can the authors elaborate their thoughts on the applications in the discussion? Could knotted structures provide more biological stability for future biomedicine applications?

We have added the discussion of possible stability: “Although knots rarely occur in proteins and DNA, and RNA knots is still elusive up to date, it is believed that knots may play important roles in various biological systems and evolutionary process, particularly during the origins of life (*Rna Biol* 13, 134-139, doi:10.1080/15476286.2015.1132069 (2016)). The entanglement may provide extra stability to resist environmental temperature changes.”

It seems when the crossing number gets very high, the yield dropped significantly. This is understandable as the strand has to weave through so many cavities in its own tanglement. It is great to see the authors used hierarchical folding pathway to improve the yield, but can the authors comment on the limitation of hierarchical folding, given the fact that they will run out of choices for sequences?

As the reviewer commented, there are limitations with hierarchical folding. We have included a statement in the discussion section.

Are the authors limited to only knotted structures? Is it possible to create other interesting topological structures, such as catenanes?

The knotted ssDNA origami strategy can be applied to form catenanes. We have included a statement in the discussion section.

Reviewer #2 (Remarks to the Author):

The work by Qi et al. on “Programming Molecular Topologies from Single-stranded Nucleic Acids” demonstrates the design and experimental methods to form 2D and 3D nanostructures from a single strand of DNA or RNA. The work is closely related to recent work by the same labs published in 2017 (ref 37). The work is in itself impressive and well described, but the question is what advances have been achieved compared to the prior work on single stranded DNA nanostructures?

The two most significant differences are

- 1) Contrary to the former work the structures in the current MS contains knotted topological features. As a consequence, the folding pathway becomes highly important and it was demonstrated that good to high yields of folded structures are obtained by the right design.
- 2) The method was used to fold 3D wireframe structures whereas only 2D structures were formed in prior work.

These are significant and important advances and therefore I do believe that the work has sufficient impact to justify publication in Nature Comm. The manuscript is well-written and the work well-documented. I am however not sure if it is necessary to include the section on the synthesis of the long ssDNA/RNA unless it differs significantly from the method described in ref 37.

We believe it is necessary to include this section.